# Automating High-Quality Concept Banks: Leveraging LLMs and Multimodal Evaluation Metrics

## Abstract

Interpretablility in recent deep learning models has become an epicenter of research particularly in sensitive domains such as healthcare, and finance. Concept bottleneck models have emerged as a promising approach for achieving transparency and interpretability by leveraging a set of human-understandable concepts as an intermediate representation before the prediction layer. However, manual concept annotation is discouraged due to the time and effort involved. Our work explores the potential of large language models (LLMs) for gener- ating high-quality concept banks and proposes a multimodal evaluation metric to assess the quality of generated concepts. We investigate three key research questions: the ability of LLMs to generate concept banks comparable to existing knowledge bases like ConceptNet, the sufficiency of unimodal text-based seman- tic similarity for evaluating concept-class label associations, and the effectiveness of multimodal information in quantifying concept generation quality compared to unimodal concept-label semantic similarity. Our findings reveal that multimodal models outperform unimodal approaches in capturing concept-class label simi- larity. Furthermore, our generated concepts for the CIFAR-10 and CIFAR-100 datasets surpass those obtained from ConceptNet and the baseline comparison, demonstrating the standalone capability of LLMs in generating high-quality con- cepts. Being able to automatically generate and evaluate high-quality concepts will enable researchers to quickly adapt and iterate to a newer dataset with little to no effort before they can feed that into concept bottleneck models.

## 1 Introduction

In recent years, large-scale machine learning models, particularly deep neural networks, have achieved remarkable improvements in accuracy across various domains. However, these advancements have often come at the expense of interpretability and transparency, making it challenging to understand the internal decision-making processes of these models. This lack of clarity and inter- pretability poses significant limitations to their deployment in critical areas where the consequences of incorrect predictions can be severe. In domains such as medical diagnostics, healthcare, public infrastructure safety, and visual inspection for civil engineering, the ability to explain and justify the decisions made by these models is of utmost importance (Li et al., 2022b). Stakeholders in these fields require a clear understanding of the reasoning behind the model's predictions to ensure that the outcomes align with established domain knowledge and best practices. Without this level of transparency, the trustworthiness and reliability of these models come into question, hindering their widespread adoption in safety-critical applications (Gao & Guan, 2023).

To address this challenge, researchers and practitioners are actively exploring methods to enhance the interpretability of machine learning models while maintaining their impressive performance (Singh et al., 2020). Techniques such as feature importance analysis, SHAP, rule extraction, and vi- sual explanations aim to provide insights into the factors influencing the model's predictions (Zhang et al., 2021; Bujwid & Sullivan, 2021; Lundberg & Lee, 2017). By bridging the gap between the model's internal workings and human understanding, these approaches seek to achieve greater con- fidence in the use of machine learning in high-stakes scenarios.

Concept Bottleneck Models (CBMs) is one of the techniques to have gained significant attention in the field of Artificial Intelligence due to their ability to provide interpretable explanations for model predictions. Just before the classification layer, Concept-Bottleneck models have a bottleneck layer that comprise of human-interpretable concepts (Oikarinen et al., 2023; Yang et al., 2023). Concept Activation Vectors (CAVs) also provide human-friendly interpretation of the existing classification models (Kim et al., 2018a). A recent modification of CBM known as Counterfactual CBM is proposed which uses counterfactual explanations by emphasizing not only on "why" but also on "what if" by providing alternate counterfactuals concepts (Dominici et al., 2024). While aforementioned approaches are promising and highly interpretable, they suffer from two major challenges: 1) Concept generation turns out to be a key challenge in concept-bottleneck models as high number of related concepts generally tend to produce better bottleneck layer resulting in more interpretability; 2) There is limited literature on independently evaluating, and hence improving concept quality before feeding them to the CBM pipeline.

In order to overcome the aforementioned challenges, there have been attempts to automate concept generation and quality improvement. For generating high quality concepts, earlier methods have mostly relied on manual concept annotation (Koh et al., 2020). While this method may generate concepts of reasonable quality, it has huge resource limitations and relies completely on human understanding of the underlying classes. Moreover, this method cannot be generalized across newer datasets as new class labels will require concept annotation from scratch. To alleviate this problem, some researchers have also proposed the idea of augmenting the base human-labelled concepts by using LLMs via in-context learning (Tan et al., 2024). A few researchers have also focused on leveraging concept annotations in datasets where it is readily available and utilise multimodal models to learn or discover new set of concepts (Wang et al., 2023; Yuksekgonul et al., 2022). Recent advancements in Large Language Models (LLMs) have shown promising results in various natural language processing tasks (Kojima et al., 2022). LLMs, such as GPT-3 have the ability to generate coherent and meaningful text based on a given prompt. This capability can be leveraged to automate the concept generation process in CBMs, reducing the manual effort required and improving the overall efficiency of the model (Yang et al., 2023).

While concept generation has been automated by the use of pre-trained LLMs, the quality assessment of the generated concepts still remains to be a challenge. Current approaches rely on running end-to-end pipeline for CBM in order to assess the quality of the generated concepts. Higher scores in CBM classification predictions are automatically interpreted as a generating good concepts (Oikarinen et al., 2023; Yuksekgonul et al., 2022). This approach does not directly quantify the quality of generated concepts. Moreover, it also requires extensive resources as running complete end-to-end pipeline is not computationally inexpensive.

In this research, we propose an unsupervised concept generation and evaluation technique which could help evaluate and iterate on the generated concepts at an early stage of CBM classification. Our method aims to eliminate the reliance on manual annotation and improving the interpretability of CBMs. We also evaluate text-based model for concept quality evaluation to see how well can it quantify the overall concept quality. Our work is inspired by some of the approaches that have achieved success in similar tasks (Semenov et al., 2024; Bhaskar et al., 2017; Kritharoula et al., 2023).

We emphasize on the following three Research Questions in the given research:

- **RQ1:** Can large language models without visual information generate good enough concept bank as compared to the existing knowledge bases such as ConceptNet?

- **RQ2:** Is unimodal text-based semantic similarity enough to evaluate the association between concepts and class labels?

- **RQ3:** Is multimodal information enough to quantify the quality of the concept generation as opposed to the unimodal concept-labels semantic similarity?

## 2 LITERATURE REVIEW

Concept Bottleneck models have been used to add a layer of interpretability to the black-box deep learning-based classification algorithms. However their performance comparatively remains to be

limited due to the lack of good quality concepts. There are various approaches for improving performance of concept-bottleneck models.

Early approaches rely heavily on manually hand-crafted concepts which may result in good human interpretation but are not scalable and require manual labor (Koh et al., 2020). Moreover, the concepts are limited to human's capability and understanding of the domain and are likely to miss some important concepts (Shang et al., 2024). Iterating and evaluating these concepts requires more manual intervention which becomes infeasible due to the lack of time and resources. Additionally, manual concept generation is subjective and the quality of concepts may rely on the individual brilliance of the annotators.

A few researchers propose methods to partially eliminate the manual annotation by proposing to learn concepts from the dataset. For instance, (Wang et al., 2023) uses self-supervision to simultaneously learn concepts and classification objective. It uses slot attention-based mechanism to spot the region where corresponding concept is found. It learns a set of k concepts while k is a hyper-parameter and is set to 5. Their experiments demonstrate that contrastive loss with self-supervision really contribute to concept discovery. The authors evaluate the accuracy of their proposed approach by generating synthetic dataset proposed in . There is one limitation of this approach and is associated with tuning the number of k-concepts for each dataset which hurts the scalability across datasets.

Another approach relies on base high-quality human annotated concepts to create a seed concept bank. These concepts are incrementally bootstrapped by learning and optimizing learnable vectors initalized from multimodal model such as CLIP (Radford et al., 2021). These ambiguous and unclear vectors are then translated into potentially meaningful concepts by using concept discovery module. Lastly, they introduce a metric to evaluate concept utilisation efficiency (Shang et al., 2024). While their approach seems to marginally outperform existing models, it relies on high quality initial seed concept bank which requires manual effort.

Hierarchical concept learning has also been explored to improve the performance of concept-bottleneck models by aiming to produce better concepts (Sun et al., 2024). The idea here is to avoid information leakage issue by introducing supervised learning in concept prediction. The authors establish notable improvements in model performance as concept prediction results in better concepts.

Due to the widening popularity of LLMs for unsupervised learning, recent articles have also dived deeper into utilising them for concept generation. A recent study Oikarinen et al. (2023) proposed a label-free concept bottleneck mechanism to generate models using GPT-3 model. They also filter concepts by utilising vision and text encoders to compute similarities between concepts and classes. A similar approach is proposed by Yang et al. (2023) where the concepts are generated automatically by LLM. However, concept filtering is performed by using submodular optimization which tends to be more effective compared to the static rules applied in the former approach. While these methods do alleviate the manual effort of generating concepts, they rely on a paid GPT-3 API. These approaches also evaluate concept quality based solely on the final results of classification which is resource intensive. Moreover, their models do not outperform existing model such as Standard sparse model on CUB-200 dataset.

Another popular approach is the use of TCAV (Testing with Concept Activation Vectors) for interpreting neural network decisions, the researchers demonstrated several key findings (Kim et al., 2018b). TCAV provides a human-friendly linear interpretation of deep learning models, offering insights into model decisions through natural high-level concepts that do not need to be predefined during training. The approach supports accessibility, customization, plug-in readiness, and global quantification, making it a versatile tool that requires minimal machine learning expertise to employ. Unfortunately, the assumption of linearity between concept and predictions does not always hold true resulting in performance degradation where non-linear dependencies need to be captured.

ConceptSHAP improves the assessment of concept importance in model explanations by adapting Shapley values to fairly assign the importance of each concept (Yeh et al., 2020). This adaptation allows it to uniquely satisfy desired axioms such as efficiency, symmetry, dummy, and additivity. Specifically, ConceptSHAP measures how much each individual concept contributes to the overall completeness score of the model, which helps in evaluating the importance of each discovered concept in explaining the model's decisions. By providing both global attribution and per-class

saliency, ConceptSHAP offers a more nuanced and interpretable understanding of how different concepts contribute to model predictions. This approach is validated through metrics and user studies on synthetic and real-world datasets, demonstrating its effectiveness in finding complete and interpretable concept explanations.

Multimodal models have also shown great improvements in tasks that involve multiple modalities such as vision and text (Li et al., 2022a). The idea of using multimodal models for concept annotation by leveraging multimodal models to obtain concept representations is also explored (Yuksekgonul et al., 2022). This involves learning the concept bank by training a linear SVM for each concept. The vector normal to the boundary is used to represent the concept. Multimodal model (CLIP Radford et al. (2021)) is then used to map each concept to a vector using text encoder. This method has a number of limitations. Firstly, it necessitates the creation of a predefined library of initial concepts, which may involve concept pruning, requiring human intervention via annotation. Secondly, this approach relies on preexisting knowledge graphs, such as ConceptNet (Speer et al., 2017), to identify relationships between classes and related concepts. While these knowledge graphs provide valuable information, they may not always capture the nuances and context-specific relationships present in the particular task or dataset at hand. Lastly, it also requires supervised training of a classifier the tuning of which also requires additional effort when adapting to a newer dataset. Consequently, the effectiveness of this approach may be limited by the quality and relevance of the utilized knowledge graphs.

A recent approach proposes a hypothesis for quantifying concept similarity using an algorithm called Concept Matrix Search (CMS) algorithm (Semenov et al., 2024). It generates concepts using ConceptNet, a popular freely-available commonsense knowledgebase, and utilises CLIP model for computing concept-image and concept-labels similarity matrices. It predicts the class label by using cosine similarity for k-th image-concept and image-class concept. While the hypothesis seems reasonable, it relies on the fact that textual embeddings for class-concept have closer semantic adherence to the image-concept mapping which may not always be true due to the abstract nature of the concepts. For example, the distance between an image of class label "apple" and the image of a red apple may be closely associated while the semantics between the label "apple" and concept "red" may not be close enough in the embeddings space. Moreover, ConceptNet is does not provide comprehensive concepts for multi-word phrase classes.

In conclusion, various approaches have been proposed to improve the performance of concept-bottleneck models. These methods aim to enhance concept discovery, reduce information leakage, and automate concept generation. While some approaches, such as self-supervision and hierarchical concept learning, have shown notable improvements in model performance, others, like label-free concept bottleneck mechanisms using GPT-3, still face challenges in terms of concept quality and cost-effectiveness. Additionally, methods that rely on predefined concept libraries and knowledge graphs may be limited by the relevance, accuracy, and completeness of these resources. Although there are developments to this field, there remains room for further research and development in this area to address the limitations and improve the scalability and generalizability of concept-bottleneck models across different datasets and tasks.

## 3 METHODOLOGY & IMPLEMENTATION

To answer the aforementioned research questions, we need to establish benchmark datasets to quantify and compare the quality of two or more concept generation sources. Once we obtain these sets of concepts against respective classes, we can apply the proposed metric to compare and evaluate their quality. Our core methodology tends to be completely unsupervised as generate we rely on LLMs for concept set generation. For concept quality evaluation, we propose a metric that relies on exploiting the pretrained knowledge of the multimodal model like CLIP for making the predictions. This results in a "training-free" methodology which may help scale to any dataset without the need for additional training and dataset-specific manual labelling. The proposed approach will help us quantify the quality of concept bank and iterate over them fast before feeding them to a larger concept-bottleneck based architecture.

## 3.1 Concept Set Generation

We generate class-wise concepts using ConceptNet and recent Large Language Models. Specifically, we generate three set of concepts as listed below:

- Random concepts (via prompting)
- ConceptNet-based concepts to serve as baseline
- LLM-generated concepts

**ConceptNet-based Concepts:** We use ConceptNet API to generate relevant concept against each class. Following the footsteps of Semenov et al. (2024), we keep only the concepts having *HasA, IsA, PartOf, HasProperty, MadeOf, and AtLocation* relationships with the class labels. We generate as many concepts as possible. Due to API limitations, we use Sentence Transformer's roberta-based [1]) model and find more concepts using algorithm similar to (Oikarinen et al., 2023).

**LLM-generated Concepts:** For automated concept generation using LLMs, we prompt recent LLMs like LLaMa3-70B AI@Meta (2024) and Qwen2-72B Bai et al. (2023) using various prompts. We used special technique in prompting to generate more granular and abstract concepts. We achieve this by prompting model in the following manner:

```
{<class label>} {<is/has relationship>}
attribute/characteristic
```

This technique generates phrases ending with unique attributes of the given class label. We then parse and keep only the attribute at the end of the phrase. We also prompt model to generate a single word or two-word phrase as suggested by Shang et al. (2024) to help achieve concept utilization. These concepts are generated against each class label individually. We restrict prompt to generate at most 15 concepts as we observe that higher number of concepts may introduce redundancy. For CUB-200, we slightly modify the prompt to generate data more specific to the attributes of birds. We notice that this approach helps generate more distinguishable concepts resulting in better performance. This also underscores the significance of task-specific prompt tuning. For the sake of reproducibility, we set temperature to 0 for generation.

**Random Concepts:** In order to assess the reliability of our proposed concept evaluation metric, we also generate a random set of concepts. We prompt LLaMA3 to generate irrelevant and unrelated concepts given a class label.

All the prompts can also be found in Appendix A.1.

## 3.2 Concept Filtering

Once the concepts are generated, we retain only the diverse set of concepts without losing much novelty ensuring high quality subset of concepts. We apply filtering criteria such as length of characters per concept in order to remove unnecessarily long concepts. Specifically, we remove any concepts smaller than 3 characters and the ones larger than 32 characters. We also preprocess and remove the concepts which can be matched with class label as a subword. Table 1 below summarizes the number of concepts after filtering against each dataset.

## 3.3 Concept Quality Evaluation

### 3.3.1 Experimental Setup

In order to evaluate the proposed solution, we declare the an experimental setup which contains three set of concepts against three popular image classification datasets including CIFAR-10, CIFAR-100 Krizhevsky et al. (2009), and CUB-200. We randomly sample a set of fifty images per class from

---

[1]We use model available here: `https://huggingface.co/sentence-transformers/all-mpnet-base-v2`

Table 1: Number of unique concepts across datasets after filtering

| Dataset | Random | ConceptNet | LLM-generated |
|---------|--------|------------|---------------|
| CIFAR-10 | 457 | 223 | 180 |
| CIFAR-100 | 1240 | 1096 | 1365 |
| CUB-200 | 1020 | 869 | 1292 |

each dataset to do class-level feature representation. The goal here is to avoid using the complete dataset and only using the sample for faster pipeline iteration.

We hypothesize that if a certain concept evaluation metric is reliable, there will be no significant difference between the scores of randomly generated concepts when compared with ones generated via LLMs and ConceptNet.

### 3.3.2 BERT-SCORE BASED EVALUATION

In order to evaluate the effectiveness of a unimodal text similarity model, we use a popular metric called BERTScore (Zhang et al., 2019). BERTScore, as the name suggests is based on BERT Devlin et al. (2018) model, and has been extensively used to compute text similarity between two sentences or words. We first compute class and concept embedding matrices using BERTScore and then compute cosine similarity between both matrices. Then we find the top-k concepts against each class and match those top concepts with the ground truth. We compute the accuracy based on the number of matches and divide by the total number classes available. The results of the experiments are detailed in Table 2 of results Section.

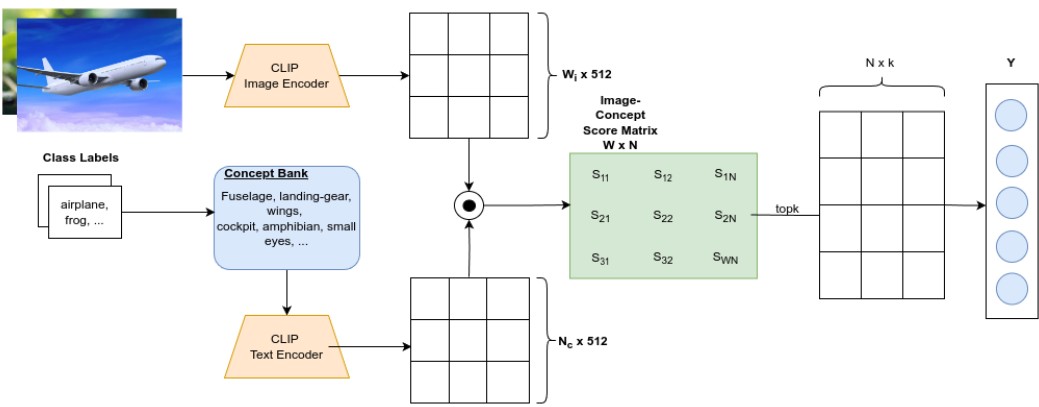

Figure 1: Proposed System Architecture Diagram

### 3.3.3 CONCEPT-DRIVEN CLASS LABEL PREDICTION USING CLIP

We propose a concept-driven class label prediction scheme that relies on multimodal features. The fundamental thought process behind the idea is to predict class label by image-concept similarities as illustrated in the Figure 1.

**Preprocessing Concepts:** Before passing the concepts to the embedder, we make the concepts unique. The goal here is to avoid redundancy in the resulting image-concept similarity matrix.

**Prefix Prompting:** We use OpenAI's CLIP model (ViT-B/32[2]) for mapping image and text embeddings to a shared embedding space. The embedder projects images and concepts into an embedding with each embedding having 512 dimensions. Before embedding concepts, we prepend a prefix

---

[2]We use openclip's implementation found here: `https://github.com/mlfoundations/open_clip`

**Algorithm 1** Concept-driven class label prediction

---

**Input:** Set of concepts $C$, Image embedding matrix $V$, Number of top concepts $k$
**Output:** Image-concept similarity matrix $M_v$, Set of top-$k$ concepts $T$, Predicted classes
$C_{unique} \leftarrow$ uniqueElements($C$) {Make concepts unique}
$C_{embedded} \leftarrow$ embedConcepts($C_{unique}$, prefix) {Embed concepts}
$M_v \leftarrow \emptyset$ {Initialize similarity matrix}
**for** each image embedding $v_i \in V$ **do**
    **for** each concept embedding $c_j \in C_{embedded}$ **do**
      $s_{ij} \leftarrow \frac{v_i \cdot c_j}{|v_i||c_j|}$ {Normalized dot product}
      $M_v[i, j] \leftarrow s_{ij}$ {Store similarity score}
    **end for**
**end for**
$T \leftarrow$ topKConcepts($M_v, k$) {Find top-$k$ concepts}
classes $\leftarrow$ matchConceptsToClasses($T$) {Match to classes}
**return** $M_v, T$, classes

---

Table 2: Comparison of Accuracy (%) by top-7 concepts using BERTScore

| Method | CIFAR10 | CIFAR100 | CUB200 |
|---|---|---|---|
| Random concepts | 20.0 | 7.0 | 6.0 |
| ConceptNet concepts | **60.0** | **24.0** | 10.0 |
| LLM-generated concepts | 50.0 | 21.0 | **11.0** |

to see if they impact the image-concept similarity score. We experiment with different prefixes as given below:

- The object in the image is/has {concept}

- The object in image comprises of {concept}

- The object in the image features {concept}

To our surprise, the prefix tuning results in higher scores as compared to embedding concept without any prefix. This also significantly impacts the semantic similarity scores, hence resulting in the final results.

We represent image embedding matrix with $V$ and concepts matrix with $C$. We find dot product between sampled images and all the concepts $M_v$ and normalize by the dot product of their norms. Now $M_v$ is a matrix containing similarity scores between image-concept similarities. We find top-k concepts with highest similarity across images. The top concepts are then matched back to find the right class. It must be noted that our approach also helps with concepts matching if they are being shared across multiple classes which means that a concept appearing in multiple classes can be used to predict all of those classes. The results of the experiments using Algorithm 2 are reported in Section 4.

## 4 EVALUATION AND DISCUSSION

We report the results against the three main datasets including CIFAR-10, CIFAR-100, and CUB-200 as mandated in earlier sections. For **RQ2**, we report the results achieved via BERTScore against the three set of concepts from different source including Random, ConceptNet, and LLM-generated concepts in Table 2. We choose the value of k as 7 for the top-k concepts. As evident from the table, there is no significant difference between the scores for CUB-200 dataset across three set of concepts. For CIFAR-100 dataset, we can also observe that the results are poor. This gives an idea of fundamental lack of understanding between concepts and class labels.

In order to delve into **RQ3**, we asses the proposed concept-driven multimodal CLIP-based approach over the same experimental setup. We also compare our results with Semenov et al. (2024) which is

Table 3: Accuracy (%) comparison against top-7 concepts using proposed approach

| Method | CIFAR10 | CIFAR100 | CUB200 |
|---|---|---|---|
| Random concepts | 16.40 | 3.78 | 6.15 |
| ConceptNet concepts | 89.40 | 55.90 | 42.02 |
| CMS (baseline) (Semenov et al., 2024) | 85.03 | 62.95 | **67.17** |
| **LLM + CLIP (Ours)** | **98.20** | **64.06** | 34.40 |

similar to our proposed technique as they also propose a training-free system for concept evaluation. Our results exhibit superior performance to their methodology across two datasets: CIFAR10 and CIFAR100. However, our approach lags behind against theirs in CUB-200 dataset. For **RQ2**, we can see that there is a huge disparity between the scores of randomly generated concepts and the concepts we generated via LLM in the results table. This outlines the reliability of our proposed evaluation metric. We further provide supportive evidence by plotting the similarity scores against the samples from CIFAR-100 and CUB-200 predictions in the Appendix A.3. Moreover, our LLM-generated concepts also outperform ConceptNet-based concepts in all three datasets showcasing the superiority of the LLMs as opposed to a knowledge based such as ConceptNet reflecting on **RQ1**.

We also evaluate our proposed technique over different values of k for top-k concepts across all datasets. The results can be seen in Figure A.2, 2 and 3 of the Appendix. We can see that the trend of accuracy is monotonically increasing as we increase the value of k. However, the statistical significance of results will drop drastically as we go beyond the value of 7 as we have maintained an average number of concepts per class to be 15.

METHODOLOGY FOR IDENTIFYING TOP-CONCEPTS ACROSS CLASSES

In order to identify the most representative concepts for each class, we employ a multi-step approach that leverages classwise similarity scores between images and concepts. The process is outlined as follows:

1. **Classwise Similarity Score Computation**: For each class, we calculate the similarity scores between the images belonging to that class and the entire set of concepts.

2. **Classwise Mode Determination**: Once the classwise similarity scores are obtained, we determine the mode (most frequently occurring value) for each class in order to get the highest level of similarity across the images within a particular class.

3. **Top-k Concept Selection**: Based on the classwise modes, we select the top-$k$ concepts for each class corresponding their mode values.

4. **Concept Retrieval**: Finally, we retrieve the concepts associated with the top-$k$ mode values for each class.

By following this methodology, we are able to effectively identify the top-concepts across classes, providing a concise and meaningful representation of the visual content within each class. This approach enables us to gain insights into the key concepts that are statistically most relevant and discriminative for each class, facilitating further analysis and understanding of the underlying data.

The identification of top concepts across classes plays a crucial role in various applications, such as image classification, retrieval, and understanding. By focusing on the most representative concepts for each class, we can develop more efficient and accurate models that capture the essential characteristics of the visual data, ultimately leading to improved performance in downstream tasks. Top concepts against selected class labels for CIFAR-100 and CUB-200 can be found in Table 4 and 5 respectively.

## 5 CONCLUSION

In this research we explored the potential of large language models (LLMs) for generating concept banks and evaluated the effectiveness of unimodal and multimodal approaches for assessing the qual-

---

**Algorithm 2** Identifying Top-Concepts Across Classes

**Input:** Set of images $I$, Set of concepts $C$, Number of top concepts $K$
**Output:** Set of top-concepts $T$ for each class
**for** each class $c_i \in C$ **do**
   $S_i \leftarrow \emptyset$ {Initialize similarity scores for $c_i$}
   **for** each image $I_j \in I$ belonging to class $c_i$ **do**
     **for** each concept $c_k \in C$ **do**
       $s_{jk} \leftarrow \text{Similarity}(I_j, c_k)$
       $S_i \leftarrow S_i \cup \{s_{jk}\}$
     **end for**
   **end for**
   $m_i \leftarrow \text{Mode}(S_i)$ {Mode of similarity scores}
**end for**
$M \leftarrow \{m_1, m_2, \ldots, m_{|C|}\}$ {Set of classwise modes}
$T \leftarrow \emptyset$ {Initialize set of top-concepts}
**for** each class $c_i \in C$ **do**
   $T_i \leftarrow \text{TopK}(M, K, c_i)$ {Select top-$K$ concepts}
**end for**
**return** $T$

---

ity of generated concepts. Our investigation was guided by three research questions: (RQ1) whether LLMs are capable of generating concept banks that are comparable to existing knowledge bases such as ConceptNet, (RQ2) whether unimodal text-based semantic similarity is sufficient for evaluating the association between concepts and class labels, and (RQ3) whether multimodal information can effectively quantify the quality of concept generation compared to unimodal concept-label semantic similarity. To address RQ1, we generated concepts using both ConceptNet as a baseline and LLMs through prompting techniques. For RQ2, we employed the BERTScore metric to evaluate the generated concepts based on their semantic similarity to the class labels. Moving forward, to tackle RQ3, we proposed a novel metric based on multimodal models such as CLIP and assessed the generated concepts using this approach. Our findings demonstrate that multimodal models are indeed necessary for accurately capturing the similarity between concepts and class labels, surpassing the performance of unimodal methods like BERTScore as well as the baseline. Furthermore, our generated concepts for the CIFAR-10 and CIFAR-100 datasets outperformed those obtained solely from ConceptNet, indicating the standalone ability of LLMs to generate high-quality concepts. However, it is worth noting that our generated concepts for the CUB-200 dataset did not surpass those from ConceptNet, highlighting the need for further investigation and improvement in this specific domain. The implications of our research are significant for the field of concept generation and evaluation. We have shown that LLMs possess the capability to generate concept banks that are competitive with existing knowledge bases, opening up new possibilities for automated concept generation. Moreover, our proposed multimodal metric provides a more comprehensive and effective approach for assessing the quality of generated concepts, taking into account both textual and visual information.

In conclusion, our research contributes to the understanding of concept generation using LLMs and emphasizes the importance of multimodal evaluation metrics. The findings suggest that LLMs have the potential to generate effective enough concepts, while multimodal models offer a more robust and accurate means of assessing concept quality. Future work can build upon these insights, exploring the usage of multimodal vision-text langauge models to generate better concepts for complex datasets like CUB-200 and refining multimodal evaluation metrics to enhance their performance across diverse datasets and domains.

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

# A APPENDIX

## A.1 PROMPTS

> **Relevant Concepts Prompt**
>
> Class Label: {object}
> Task: Generate a list of descriptive concepts and attributes that comprehensively characterize the physical properties, appearance, and features of the given class label. Consider wings, tail, pattern, shape, size, color, texture, material, and any other relevant physical aspects. Aim to create a rich set of concepts that could be used to create a detailed visual representation or description of the class. Provide a minimum of 15 concepts per class. Do not provide explanations, only word or two-word phrase
> Create concept in the format: `{class label} {is/has relationship} attribute/characteristic`
> Concepts: 1. 2. 3. 4. 5. 6. 7. 8.

> **CUB-200 Relevant Concepts Prompt**
>
> Task: Generate a list of descriptive concepts and attributes that comprehensively characterize the physical properties, appearance, and features of the given bird. Consider the shape, size, color, texture, pattern of the beak, wings, tail, feet and any other relevant physical aspects. Aim to create a rich set of concepts that could be used to create a detailed visual representation or description of the class. Provide a comprehensive set of 15 concepts per class. Do not provide explanations, only word or two-word phrase. Think step by step and then write the descriptions
> Create concept in the format: `{class label} {is/has relationship} attribute/characteristic`
> Concepts: 1. 2. 3. 4. 5. 6. 7. 8.

## A.2 TOP-K HITS PLOTS FOR DIFFERENT VALUES OF K

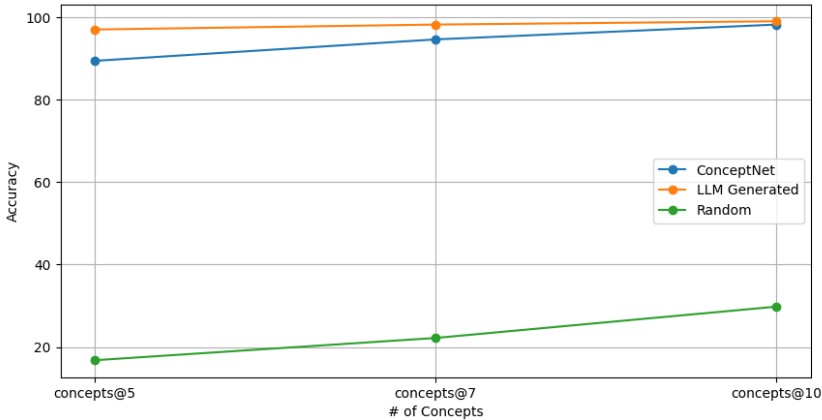

## A.3 RESULTS INTERPRETATION

We analyze the top concept scores against the misclassifications for the proposed methodology. We pick random samples from both CIFAR-100 and CUB-200 dataset. We observe that majority of the concepts match well with the image. The reason, hence, for the misclassification can be attributed for the lack of presence of those concepts in the LLM-generated ground truth. Editing the ground-truth can fix these classification mistakes.

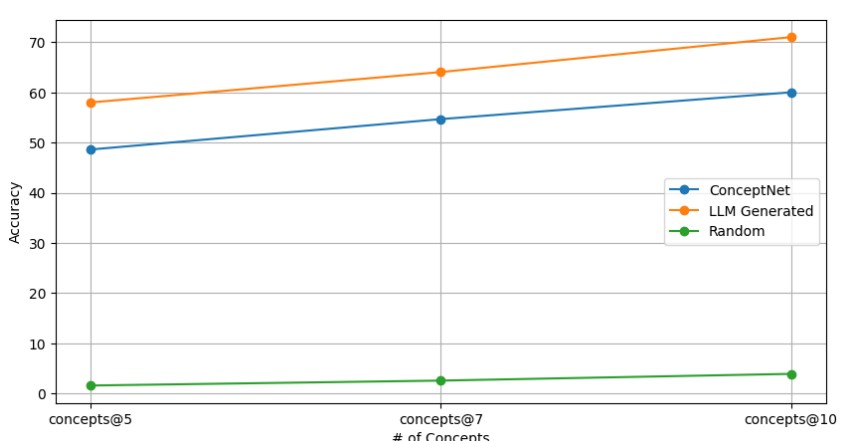

Figure 2: CIFAR100 - Accuracy results for different values of top-k

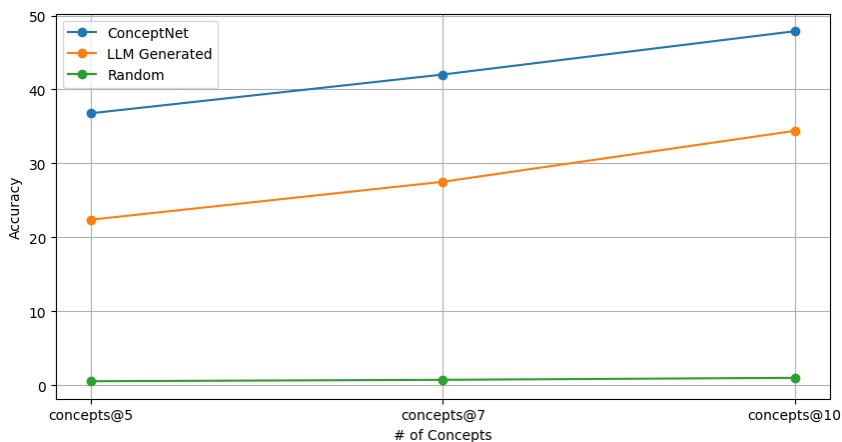

Figure 3: CUB200 - Accuracy results for different values of top-k

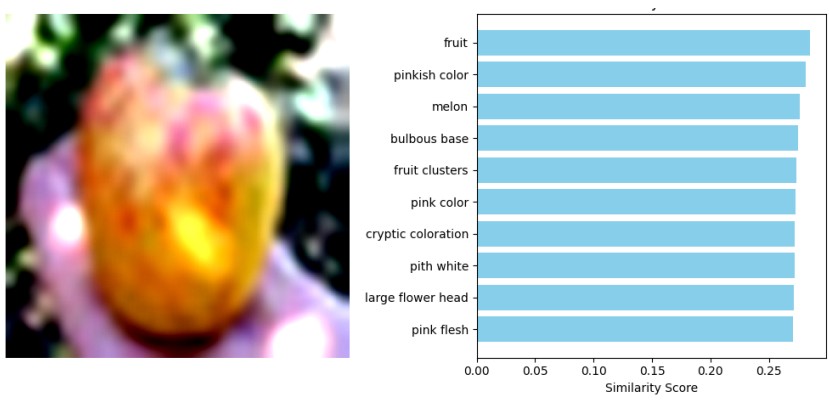

Figure 4: Top concepts for misclassified Apple

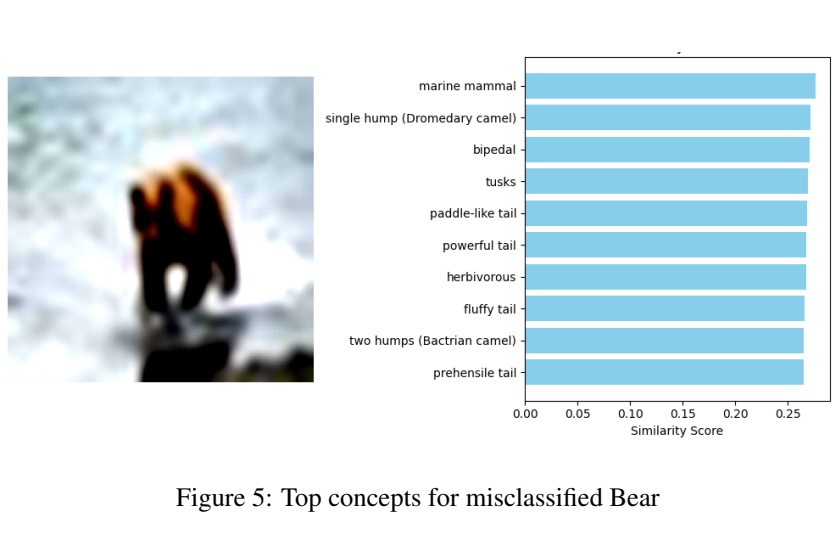

Figure 5: Top concepts for misclassified Bear

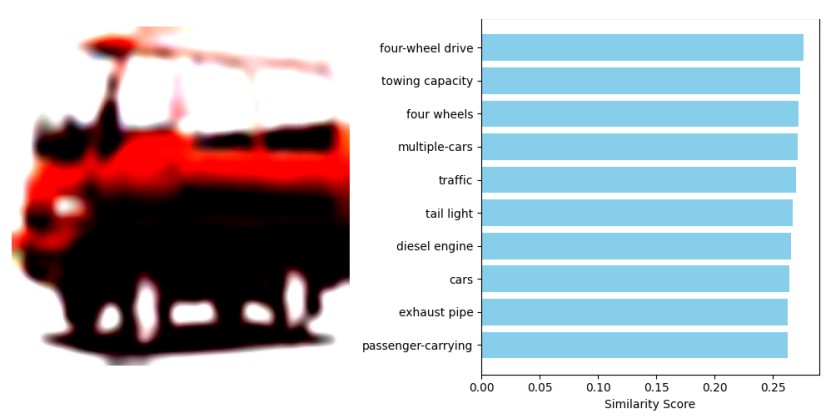

Figure 6: Top concepts for misclassified Bus

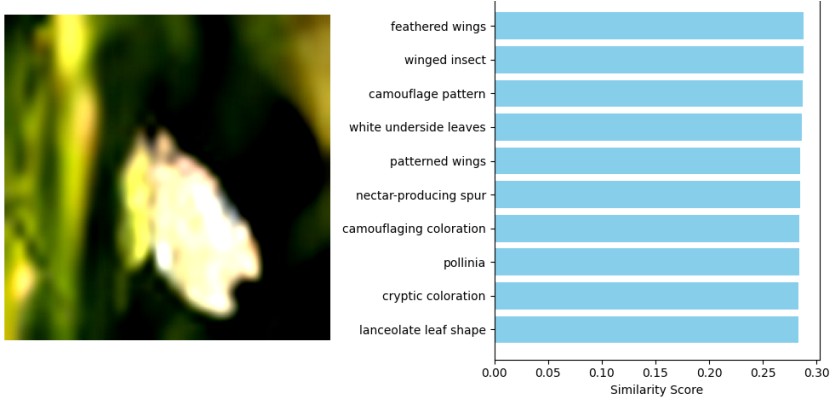

Figure 7: Top concepts for misclassified Butterfly

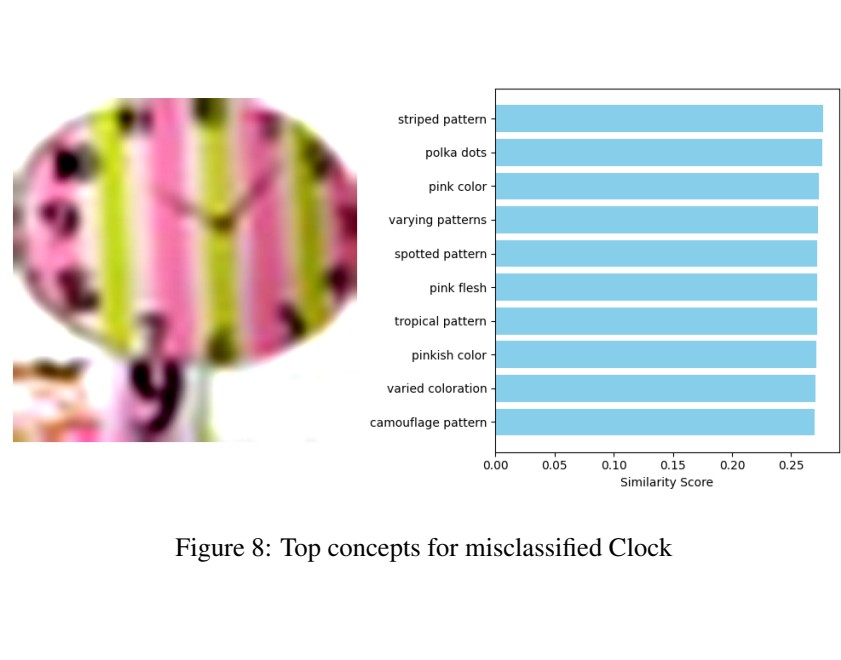

Figure 8: Top concepts for misclassified Clock

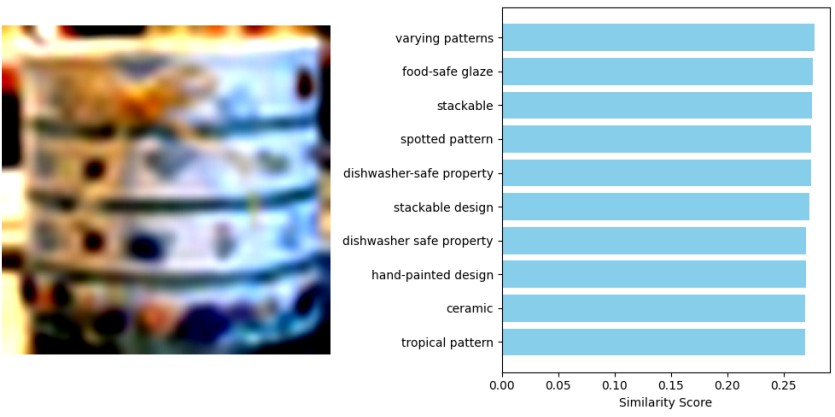

Figure 9: Top concepts for misclassified Can

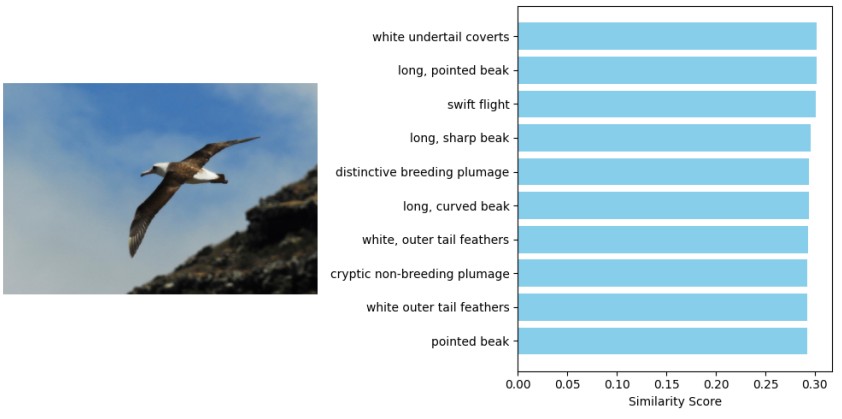

Figure 10: Top concepts for misclassified Lysan Albatross

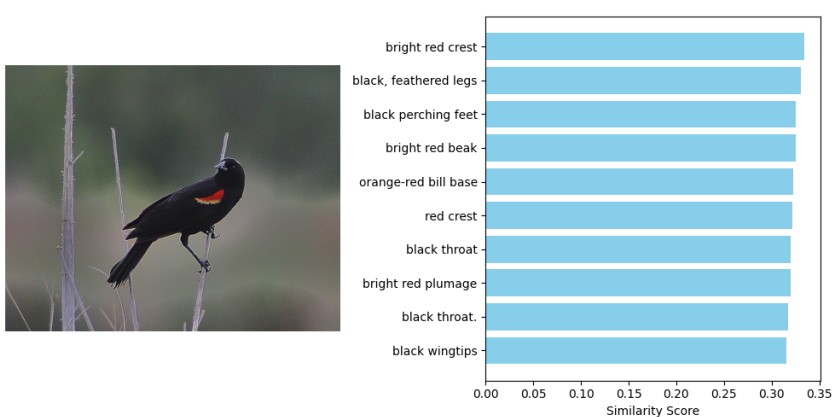

Figure 11: Top concepts for misclassified Red-winged Blackbird

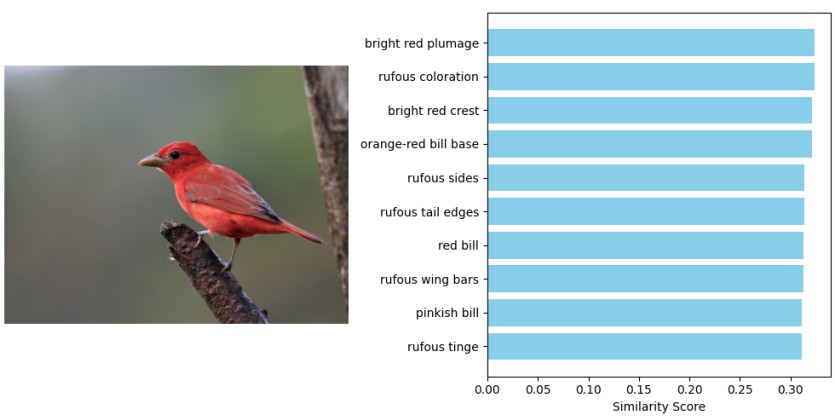

Figure 12: Top concepts for misclassified Summer Tanager

Table 4: Top Concepts for selected classes in CIFAR-100 dataset

| Class Labels | ConceptNet | LLM-generated |
|---|---|---|
| Tulip | flower, flowering plant, sweet smelling flower, garden pollinating flowers | bell-shaped flower, anthers, shallow depth, spring bloom, diverse flora |
| Tiger | big cat, felid, wild animal, zoo, pack animal, mammal | sharp claw, claws, long ears, mammal, distinctive roar |
| Whale | cetacean, marine animal, tail fin, sea world, ocean | large dorsal fin, prominent dorsal fin, caudal fin |
| Snake | snake pit, cutworms, chain, long tail, tusks | annular markings, scaly tail, spiral shape, scaly texture, cord, winding-key |
| Maple Tree | trees, angiospermous tree, forest, group of trees | orange fall color, yellow fall tree, red oak variety, deciduous leaves, tall trees |
| Chair | furniture item, table, table cloth | armrests, backrest, saddle seat, seating arrangement, adjustable height |

Table 5: Top Concepts for selected classes in CUB-200 dataset

| Class Labels | ConceptNet | LLM-generated |
|---|---|---|
| Lazuli Bunting | small common songbird, migratory bird, corvine bird, passerine, columbiform bird, finch | greenish-blue bill, vivid blue plumage, rufous sides, rufous coloration, orange bill, bright red crest |
| Cardinal | corvine bird, columbiform bird, finch, bird genus, migratory bird, piciform bird, small common songbird | bright red plumage, bright red beak, red plumage, red bill tip, bright red crest, orange-red bill base |
| Red-faced cormorant | podicipitiform seabird, gaviiform seabird, pelecaniform seabird, shore bird, sea duck, bird genus | black perching feet, feathered legs, distinctive breeding plumage, black webbed feet, black beak, glossy black plumage |
| American Crow | corvine bird, New World blackbird, columbiform bird, thrush, piciform bird, apodiform bird | annular markings, scaly tail, spiral shape, scaly texture, cord, winding-key |
| Black-billed Cuckoo | New World flycatcher, columbiform bird, New World warbler, corvine bird, migratory bird, finch | white throat, white throat patch, yellow throat, white, rufous-edged tail, rufous undertail coverts, rufous tinge, rufous sides |
| Purple Finch | small common songbird, finch, piciform bird, apodiform bird, corvine bird, bird genus, columbiform bird | pinkish bill, pinkish bill base, bright red crest, red bill, bright red beak |

