# OpenReview forum: "Automating High-Quality Concept Banks: Leveraging LLMs and Multimodal Evaluation Metrics"
_ICLR.cc/2025/Conference — Submitted to ICLR 2025_

### Official Review · Reviewer_Sv61 · 2024-11-02

**Soundness:** 2
**Presentation:** 2
**Contribution:** 2
**Rating:** 3
**Confidence:** 4

**Summary:**

The paper proposes using large language models (LLMs) to automate high-quality concept bank generation, addressing challenges in interpretability for concept bottleneck models (CBMs). A multimodal evaluation metric, leveraging models like CLIP, is introduced to assess the quality of LLM-generated concepts.

**Strengths:**

Provides a approach to concept generation, reducing reliance on manual annotation.
Introduces an unsupervised, training-free evaluation metric based on multimodal similarity, which enhances scalability.

**Weaknesses:**

The study shows limited performance for the CUB-200 dataset, with results not consistently surpassing ConceptNet, indicating potential dataset-specific constraints.

The filtering approach for concept generation lacks rigorous justification, particularly for character limits, which may impact concept quality.


Experimental comparisons lack statistical rigor, making it difficult to assess the robustness of the performance differences observed.

**Questions:**

see weakness

---

### Official Review · Reviewer_1eX4 · 2024-11-04

**Soundness:** 2
**Presentation:** 1
**Contribution:** 2
**Rating:** 3
**Confidence:** 5

**Summary:**

This paper proposes a multimodal evaluation metric to assess the quality of concepts for concept bottleneck models and explores three research questions: the ability of LLMs to generate concept banks comparable to existing knowledge bases, the sufficiency of unimodal text-based semantic similarity for evaluating concept-class label associations, and the effectiveness of multimodal information in quantifying concept generation quality. The results show that LLMs can generate high-quality concepts, particularly for the CIFAR-10 and CIFAR-100 datasets, outperforming ConceptNet and a baseline method.

**Strengths:**

- A new method that doesn't need to evaluate the quality of concept from end to end.
- Three key problems are proposed and experimental exploration is carried out

**Weaknesses:**

- I don't think that simply prompting model to generate concepts is an innovative approach, and there has been work [1] using large language models for iterative concept generation.
- There are two formatting errors in the abstract section alone: "gener- ating" and "seman- tic".
- Three of the nine pages of the text are related to work, resulting in less introduction of the author's own work.

[1]. Yue Yang, Artemis Panagopoulou, Shenghao Zhou, Daniel Jin, Chris Callison-Burch, and Mark Yatskar. Language in a bottle: Language model guided concept bottlenecks for interpretable image classification. In Proceedings of the IEEE/CVF Conference on Computer Vision and Pattern Recognition, pp. 19187–19197, 2023.

**Questions:**

- What is the innovation of using prompt technology to prompt LLMs to generate concepts  compared to this work that also uses LLMs and designs a iterative algorithm?

[1]. Yue Yang, Artemis Panagopoulou, Shenghao Zhou, Daniel Jin, Chris Callison-Burch, and Mark Yatskar. Language in a bottle: Language model guided concept bottlenecks for interpretable image classification. In Proceedings of the IEEE/CVF Conference on Computer Vision and Pattern Recognition, pp. 19187–19197, 2023.

---

### Official Review · Reviewer_aW94 · 2024-11-04

**Soundness:** 2
**Presentation:** 1
**Contribution:** 1
**Rating:** 3
**Confidence:** 5

**Summary:**

The paper explores a method to automate concept generation for Concept Bottleneck Models (CBMs) by leveraging Large Language Models (LLMs) and multimodal evaluation metrics. It proposes using LLMs to create high-quality "concept banks" that can support interpretable machine learning, especially in critical domains where transparency is crucial. Three main research questions address (1) the effectiveness of LLMs in generating concept banks, (2) whether unimodal text-based semantic similarity suffices for evaluating concept quality, and (3) the value of multimodal approaches, specifically CLIP, for enhanced evaluation.

**Strengths:**

1. the problem of generating high-quality concepts is significant in the community.

**Weaknesses:**

1. The novelty of this work is limited. The method does not significantly advance beyond prior LLM-based generation techniques, especially considering that other studies have explored similar applications of LLMs for concept generation [1] [2].

2. The proposed evaluation relies on CLIP for multimodal embedding similarity, which might introduce biases, especially as CLIP-trained models may not fully capture domain-specific subtleties in specialized fields and may have robustness issues [3]. This limitation is also evident in the lower performance on the CUB-200 dataset compared to ConceptNet-based approaches, suggesting that LLM-generated concept banks might require additional fine-tuning for more nuanced datasets.

3. Extensive LLM-based prompting and multimodal filtering may pose scalability issues, especially in resource-limited environments. However, the paper lacks a discussion on the computational costs or efficiency of generating and filtering concepts at scale.

4. The analysis provided is minimal. I recommend the authors offer more insight, such as explaining the choice of BERTScore for evaluation and detailing how text and image alignment or consistency is assessed.

5. The experiments appear incomplete, with no ablation studies to validate the choices made. Some aspects of Algorithm 1 and Algorithm 2, like the use of top-k selection and the class-matching process, lack explanation and add to the confusion. More detailed justifications and clarity on these algorithms would improve comprehensibility.

6. The paper’s organization and writing need improvement to enhance readability and coherence.

*Reference*:

[1] Oikarinen, Tuomas, et al. "Label-free Concept Bottleneck Models." International Conference on Learning Representations. 2023.

[2] Yuksekgonul, Mert, Maggie Wang, and James Zou. "Post-hoc Concept Bottleneck Models." International Conference on Learning Representations. 2023.

[3] Lai, S., Hu, L., Wang, J., Berti-Equille, L., & Wang, D. (2023). Faithful vision-language interpretation via concept bottleneck models. International Conference on Learning Representations. 2024.

**Questions:**

1. How does this approach advance beyond previous LLM-based concept generation techniques, given that similar methods have been used in prior work? Provide some explanation and insight.

2. Why was CLIP specifically chosen as the multimodal evaluation model, and are there plans to explore alternatives that might better handle domain-specific subtleties?

3. Can you clarify the computational costs of this method? How scalable is this approach in resource-constrained environments, and have you considered optimizations to improve efficiency?

4. Why was BERTScore selected as the metric for evaluating concept quality, and how does it compare to other possible metrics for aligning text and image representations?

5. Could you provide additional explanations for certain steps in Algorithm 1 and Algorithm 2, such as the reasoning behind using top-k selection and the process for class-matching?

6. Are there any plans to include ablation studies or additional experiments to clarify the impact of different components in your method?

7. The performance on the CUB-200 dataset was lower than with ConceptNet-based approaches. What fine-tuning or additional steps could potentially improve the robustness of LLM-generated concepts for complex datasets?

---

### Official Review · Reviewer_NWrY · 2024-11-04

**Soundness:** 2
**Presentation:** 1
**Contribution:** 2
**Rating:** 3
**Confidence:** 4

**Summary:**

This work proposes an analysis of using large language models (LLMs) as automatic concept generators and employing multimodal models to measure concept alignment, in contrast to existing approaches that use knowledge-based concept banks. The authors investigate three research questions (RQs) and conduct experiments on three benchmarks: CIFAR10, CIFAR100, and CUB-200.

**Strengths:**

This work looks into an important question on automatic concept generation in concept bottleneck models, aiming to improve model interpretability.

The proposed research questions are interesting and inspiring.

**Weaknesses:**

1. Limited Novelty: This work is very similar to [1], which also uses LLMs to generate concept banks and multimodal embeddings (e.g., CLIP) to identify concepts most relevant to labels. Furthermore, existing work has already explored using LLMs as automatic concept generators, as seen in [2]. The distinctions explained in Section 2, such as using "paid GPT-3 API" in [2] v.s. LLaMa3 and Qwen2 in this paper, do not represent significant novelty.

2. Insufficient Experiments: While the authors pose three research questions, the experiments do not fully address these questions and may be overly ambitious in their claims. The study only uses one type of text embedding and one kind of multimodal models, yet asserts conclusions regarding the effectiveness of "unimodal similarity" versus "multimodal similarity." The limited scope of experiments makes these conclusions feel overstated.

3. Poor Writing and Formatting: The paper's writing is unclear and lacks focus, with issues such as an overly lengthy related works section (Section 2), lots of confusing sentences and grammatical errors (e.g., "as generate" in line 210), and formatting problems such as a missing reference in "proposed in " on line 123.

[1] Yan et al., Learning Concise and Descriptive Attributes for Visual Recognition, ICCV2023.

[2] Yang et al. Language in a bottle: Language model guided concept bottlenecks for interpretable image classification, CVPR2023.

**Questions:**

What is the specific difference between this work and [1]?

[1] Yan et al., Learning Concise and Descriptive Attributes for Visual Recognition, ICCV2023.

---

### Official Review · Reviewer_eCrJ · 2024-11-05

**Soundness:** 2
**Presentation:** 2
**Contribution:** 2
**Rating:** 5
**Confidence:** 3

**Summary:**

This study explores the potential of large language models (LLMs) to automate high-quality concept bank generation, proposing a multimodal evaluation metric to assess the quality of generated concepts. The study addresses three main research questions: comparing LLM-generated concepts with existing knowledge bases (such as ConceptNet), evaluating the effectiveness of unimodal text-based semantic similarity for concept-class associations, and examining the utility of multimodal information in quantifying concept quality. The results demonstrate that multimodal models outperform unimodal approaches in capturing concept-class label similarity. The LLM-generated concepts surpass those from ConceptNet for CIFAR-10 and CIFAR-100 datasets, showing LLM’s potential for high-quality concept generation. However, for the more complex CUB-200 dataset, the LLM-generated concepts do not outperform ConceptNet, suggesting room for improvement. The study contributes to the field of automated concept generation, underscoring the importance of multimodal evaluation metrics.

**Strengths:**

Innovative Methodology: The study introduces an unsupervised concept generation method and a CLIP-based multimodal evaluation scheme, providing a more automated and adaptable solution for concept bottleneck models.


Comprehensiveness: The experiments cover three different datasets and compare both unimodal and multimodal evaluation methods, ensuring reliability in the conclusions.


Practical Relevance: The applications in concept bottleneck models hold significant practical value, potentially improving model interpretability and generalizability.


Well-defined Research Questions: The research questions are thoughtfully designed, covering LLM’s standalone generation capacity, unimodal evaluation efficacy, and the role of multimodal evaluation in concept quality assessment.

**Weaknesses:**

Dataset Representativeness: The chosen datasets (CIFAR-10, CIFAR-100, and CUB-200) are relatively outdated and small in scale, which could limit the broader applicability and generalizability of the findings.

Reliance on Prefix Tuning: The methodology’s use of prefix tuning may introduce human-induced bias, affecting generalizability. Future studies could consider more automated prefix generation methods.

Lack of Statistical Verification: The results do not include statistical analysis (e.g., t-tests) to validate the significant differences in accuracy between LLM and ConceptNet-generated concepts, which may impact the scientific rigor and persuasiveness of the results.

Picture lack of futher explaination: Figure 1: Proposed System Architecture Diagram.You should add more detail to describe your image.

**Questions:**

Could you further elaborate on the prefix tuning process for LLM-generated concepts, particularly regarding how this could be automated for different tasks?

Has the study considered incorporating more representative datasets, to enhance the applicability of the findings?

Beyond CLIP, would you consider using additional multimodal models to further verify and enhance the reliability of the results?

---

### Meta-Review · Area_Chair_dvWx · 2024-12-16

**Metareview:**

The paper explores an automated approach to generating concept banks for concept bottleneck models using LLM and a multimodal evaluation metric. The study addresses three primary research questions: comparing LLM-generated concepts with existing knowledge bases, evaluating unimodal semantic similarity, and examining multimodal information's role in concept quality assessment (Reviewers eCrJ, NWrY, aW94). The research shows potential by demonstrating that LLM-generated concepts can outperform ConceptNet for CIFAR-10 and CIFAR-100 datasets (Reviewer eCrJ). Strengths include addressing an important problem in machine learning interpretability, proposing an innovative evaluation approach, and exploring the potential of LLMs in concept generation (Reviewers eCrJ, NWrY).

However, significant weaknesses substantially undermine the paper's contributions. Reviewers consistently highlighted limited novelty, with multiple sources pointing out that similar approaches have been explored in previous research (Reviewers NWrY, aW94, 1eX4). The methodology suffers from several critical limitations, including insufficient experimental design, lack of statistical verification, and poor presentation (Reviewers eCrJ, NWrY, aW94). The experimental results are inconsistent, particularly the lower performance on the CUB-200 dataset, which raises questions about the method's generalizability (Reviewers eCrJ, aW94). Additionally, the paper was critiqued for extensive formatting issues, unclear writing, grammatical errors, and an overly lengthy related works section that fails to clearly differentiate the current work from prior research (Reviewers NWrY, aW94, 1eX4). Consequently, the reviewers unanimously recommend rejection, with ratings consistently at "reject, not good enough" or "marginally below the acceptance threshold."

**Additional Comments On Reviewer Discussion:**

No rebuttal provided

---

### Decision · Program_Chairs · 2025-01-22

Reject